# The Fibrotic Effects of LINC00663 in Human Hepatic Stellate LX-2 Cells and in Bile Duct-Ligated Cholestasis Mice Are Mediated through the Splicing Factor 2-Fibronectin

**DOI:** 10.3390/cells12020215

**Published:** 2023-01-04

**Authors:** Yang Chu, Linan Bao, Yun Teng, Bo Yuan, Lijie Ma, Ying Liu, Hui Kang

**Affiliations:** 1Department of Laboratory Medicine, The First Affiliated Hospital of China Medical University, Shenyang 110001, China; 2Department of Laboratory Medicine, The First Affiliated Hospital, College of Medicine, Zhejiang University, Hangzhou 310003, China

**Keywords:** LINC00663, splicing factor 2, fibronectin, alternative splicing, hsa-miR-3916

## Abstract

Hepatic fibrosis can develop into cirrhosis or even cancer without active therapy at an early stage. Long non-coding RNAs (lncRNAs) have been shown to be involved in the regulation of a wide variety of important biological processes. However, lncRNA mechanism(s) involved in cholestatic liver fibrosis remain unclear. RNA sequence data of hepatic stellate cells from bile duct ligation (BDL) mice or controls were analyzed by weighted gene co-expression network analysis (WGCNA). Based on WGCNA analysis, a competing endogenous RNA network was constructed. We identified LINC00663 and evaluated its function using a panel of assays, including a wound healing assay, a dual-luciferase reporter assay, RNA binding protein immunoprecipitation and chromatin immunoprecipitation. Functional research showed that LINC00663 promoted the activation, migration and epithelial–mesenchymal transition (EMT) of LX-2 cells and liver fibrosis in BDL mice. Mechanistically, LINC00663 regulated splicing factor 2 (SF2)-fibronectin (FN) alternative splicing through the sponging of hsa-miR-3916. Moreover, forkhead box A1 (FOXA1) specifically interacted with the promoter of LINC00663. In summary, we elaborated the fibrotic effects of LINC00663 in human hepatic stellate LX-2 cells and in bile duct-ligated cholestasis mice. We established a FOXA1/LINC00663/hsa-miR-3916/SF2-FN axis that provided a potential target for the diagnosis and targeted therapy of cholestatic liver fibrosis.

## 1. Introduction

HF is a repair response of the body after chronic liver injury [1]. It has various causes and is manifested by the deposition of extracellular matrix (ECM) components [2]. Several studies have shown that in both parenchymal and cholestatic liver injury, activated hepatic stellate cells (HSCs) are the main source of ECM [3]. In normal liver, HSCs remain in a non-proliferating, quiescent state. When liver injury occurs, HSCs are activated and converted to myofibroblasts, which are characterized by the expression of α-smooth muscle actin (α-SMA) and collagen, and have proliferative, contractile, inflammatory and chemotactic effects [4,5]. Activated HSCs can produce and deposit more collagen and fibronectin (FN) in the ECM to promote HF [6]. Therefore, we will focus on the mechanism of HSC activation in cholestatic liver fibrosis. Transforming growth factor-β (TGF-β) is an effective cytokine that can activate HSCs to differentiate into myofibroblast-like cells that promote fibrosis and is therefore widely used in activation models of HSCs in vitro [7].

Fibronectin is a macromolecular glycoprotein that constitutes the extracellular matrix and is secreted by HSCs after activation [8]. The pre-mRNA of FN is thought to undergo alternative splicing to produce variants such as extra domain A-FN (EDA-FN), which is crucial in fibrosis [9]. Splicing factor 2 is a clear and found earlier predominant factor in the alternative splicing of FN pre-mRNA, and the level of SF2 expression directly affects the amount of EDA-FN produced [10,11,12].

Long non-coding RNAs (lncRNAs) are endogenous RNAs longer than 200 nucleotides that are byproducts of RNA polymerase II transcription. They exist in the cytoplasm or nucleus and lack effective open reading frames [13]. LncRNAs participate in a variety of biological processes in the body, such as regulating the proliferation, apoptosis and differentiation of cells and regulating metabolism and the immune response [14]. We speculate whether lncRNA can regulate liver fibrosis by regulating the expression of SF2-FN. 

Weighted gene co-expression network analysis (WGCNA) is a biological method for analyzing correlation patterns between different samples. Since WGCNA focuses on the correlations between the co-expression modules and the external clinical traits, not merely the differences in gene expression patterns, the results are thus more reasonable. It is used by some researchers to identify key genes for breast cancer prognosis [15,16].

Hence, the aim of this study was to screen for crucial genes related to cholestatic liver fibrosis by analyzing the microarray data of HSCs from bile duct ligation (BDL) mice using WGCNA analysis. We then constructed a LINC00663-related competing endogenous RNA (ceRNA) network and verified this in TGF-β-activated HSCs. We evaluated the function of LINC00663, a lncRNA, using functional cellular and animal experiments, and found that it can be transcriptionally activated by forkhead box A1 (FOXA1). It also acts as a ceRNA of hsa-miR-3916 to participate in the alternative splicing of FN by SF2 to regulate liver fibrosis. Hopefully, these findings will contribute to the understanding of cholestatic liver fibrosis and provide therapeutic targets for its diagnosis and treatment.

## 2. Materials and Methods

### 2.1. Data Collection and Preprocessing

A GSE34640 dataset was downloaded from the Gene Expression Omnibus (GEO) database [17]. In this study, we selected eight samples, including five quiescent mouse HSCs and three bile duct ligation-activated HSCs. Unqualified samples were inspected and rejected. Data were normalized in a limma package of R software (version 3.6.1). Probes were annotated by Affymetrix annotation files. Finally, 25% of genes were screened out to construct a co-expression network using an analysis of variance calculation.

### 2.2. WGCNA

Soft threshold power (β) was selected by a pickSoftThreshold function based on scale-free topology criteria. We used this β to create a weighted adjacency matrix. Then it was converted to a topology matrix (TOM). According to the TOM-based dissimilarity (1-TOM) measurement, genes were used as a hclust function to indicate hierarchical clustering. After hierarchical clustering, strongly linked genes were assigned to the same module. The module eigengene represents the gene expression profiles and average gene expression level of the module. Module membership (MM) represents the relationship between genes and modules. Gene significance (GS) can be considered as the relationship between a single gene and clinical information [15]. The genes of the most important modules were selected to proceed with the analysis.

### 2.3. Identification and Functional Annotation of Key Genes in Co-Expression Networks

We selected the top 5% of genes in the significant module as key genes that had high connectivity in each candidate module. These genes were submitted to the DAVID database [18,19] for Gene Ontology (GO) and Kyoto Encyclopedia of Genes and Genomes (KEGG) pathway enrichment analyses. There are three main categories of GO enrichment analysis: molecular function, biological process and cellular component. A value of *p* < 0.01 shows that the difference is statistically significant. The KEGG database can analyze the metabolic pathways and functions of intracellular genome products. A value of *p* < 0.05 indicates that the difference is statistically significant.

### 2.4. Identification of Hub Genes

The protein–protein interaction (PPI) network was established by key genes using STRING [20] and visualized by Cytoscape [21]. Then, we used the mcode [22] software to screen hub genes in the PPI network (selection criteria: mcode scores > 5, degree cut-off = 2, node score cut-off = 0.2, max depth = 100 and k-score = 2). We selected genes with a degree > 10 as hub genes.

### 2.5. Construction of ceRNA Network

We used online prediction tools, such as Targetscan [23] and miRDB [24], to predict miRNAs that could bind to target mRNAs, and RegRNA 2.0 [25] to predict lncRNAs that could bind to miRNAs. Based on the ceRNA hypothesis [26], lncRNA acts as a miRNA sponge, binds directly to miRNAs, and indirectly regulates the function of mRNAs. Therefore, lncRNAs and mRNAs negatively regulated by miRNAs were selected to construct a ceRNA network. Verification was performed on activated LX-2 cells.

### 2.6. Cell culture and LX-2 Cells Activated by TGF-β

Immortalized LX-2 (Bnbio, Beijing, China) and HEK293T cell lines were cultured in Dulbecco’s modified Eagle medium (DMEM; Gibco, Waltham, MA, USA) containing 10% fetal bovine serum (FBS; Every Green, Huzhou, China). LX-2 cells were activated with TGF-β (10 ng/mL; Sino Biological, Beijing, China) for 24 or 48 h, respectively. The cells were grown in 37 °C incubators with 5% CO_2_.

### 2.7. RNA Extraction and Quantitative Real-Time Polymerase Chain Reaction

Total RNA was extracted from cultured cells or mouse tissues using TRIzol (Invitrogen, Waltham, MA, USA). We subsequently used a Nanodrop 2000 to analyze the concentration and purity of RNA. U6 and hsa-miR-3916 levels were detected using a miRNA assay system (Takara Biotechnology Co., Ltd., Dalian, China). To detect mRNA expression, a ReverTra Ace^®^ qPCR RT Kit (Toyobo, Osaka, Japan) was used to synthesize cDNA using 1 μg total RNA with a reaction volume of 10 μL. Real-time quantitative PCR (RT-qPCR) analysis was performed using THUNDERBIRD^®^ SYBR^®^ qPCR Mix (Toyobo). To normalize the data, β-actin or U6 were used as reference genes. Each reaction was performed three times. A 2^−ΔΔCt^ method was used to calculate relative gene expression levels. Table 1 lists the primers used.

### 2.8. Western Blot Analysis

Radioimmunoprecipitation lysis buffer (Beyotime Biotechnology, Shanghai, China) containing protease inhibitor cocktail (Sigma–Aldrich, St. Louis, MI, USA) was used to extract the total protein. A bicinchoninic acid method (Beyotime Biotechnology) was used to measure the concentration of protein. Protein samples were separated by sodium dodecyl sulfate 10% or 6% polyacrylamide gel electrophoresis and transferred to a polyvinylidene difluoride transfer membrane (0.45 µm; Millipore, Burlington, MA, USA). Incubation of the blot with primary antibody was performed overnight at 4 °C after blocking. Finally, we used Pierce ECL Western blotting substrate (Millipore) to visualize the membrane. All immunoblots are n = 3 (biological replicates) and *p* < 0.05 represents statistically significant results. We used the following primary antibodies in this study: α-SMA (1:1000, CST, Danvers, MA, USA, #59), collagen type I alpha 1 chain (COL1A1; 1:500, Santa Cruz, Dallas, TX, USA, sc-59772), β-actin (1:1000, CST, #60), E-cadherin (E-cad; 1:1000, CST, #999), vimentin (1:1000, CST, #7431), desmin (1:1000, CST, #1674), SF2 (1:1000, Abcam, Cambridge, UK, ab38017), FN (1:5000, Abcam, ab45688), extra domain A-FN (EDA-FN; 1:500, Santa Cruz, sc-59826) and FOXA1 (1:5000, Abcam, ab170933).

### 2.9. Cell Transfection

Small interfering (si)-LINC00663, pcDNA-LINC00663, si-FOXA1, pEGFP- FOXA1, hsa-miR-3916 mimic, hsa-miR-3916 inhibitor and their negative controls were all designed and synthesized by GenePharma (Suzhou, China). Cells were transfected with si-LINC00663 (SI-LIN-1 and SI-LIN-2), si-FOXA1 (SI-FOXA1-1 and SI-FOXA1-2) and their control (SI-NC) at a final concentration of 80 nM. The cDNA of LINC00663 (pcDNA-LIN) and FOXA1 were cloned into pcDNA3.1 and pEGFP vectors, respectively. Cells were transiently transfected with a mimic or inhibitor of hsa-miR-3916 and their negative control (miR-NC) using a final concentration of 50 nM and 150 nM, respectively. To perform cell transfection, we used Lipofectamine 3000 (Invitrogen, Waltham, MA, USA) according to the manufacturer’s protocol. The above sequences are shown in Appendix A.

### 2.10. Dual-Luciferase Reporter Assay

In the LINC00663 target hsa-miR-3916 and hsa-miR-3916 target gene SF2 luciferase reporter assay, LX-2 cells were cotransfected with pmirGLO report vectors containing wild-type (WT) LINC00663, mutant (MUT) LINC00663 or WT-SF2, MUT-SF2 and hsa-miR-3916 mimics or miR-NC. In the FOXA1 target LINC00663 luciferase reporter assay, HEK293T cells were cotransfected with a pmirGLO report vectors containing WT-LINC00663, LINC00663 promoter region mutation and FOXA1 or NC plasmids. Forty-eight hours after transfection, we used a dual luciferase reporter assay system (Promega, Madison, WI, USA) to measure luciferase activity. Using Renilla luciferase activity as a baseline, the data were normalized.

### 2.11. RNA Fluorescence In Situ Hybridization Assay

5′ fam-labeled LINC00663 probes (GenePharma) were used for RNA fluorescence in situ hybridization (FISH) assays in LX-2 cells according to the manufacturer’s protocol. The probe sequence was 5′-CGTTCTTTCACGTCCCTAGCTGTATTCACTCTCCCTG-3′. Confocal images were captured using a Nikon confocal microscope.

### 2.12. RNA Immunoprecipitation

We performed RNA immunoprecipitation (RIP) experiments using a Magna RIP^TM^ RNA-Binding Protein Immunoprecipitation Kit (Millipore) according to the manufacturer’s protocol. The antibody used for RIP was 5 μg of anti-Ago2 (Millipore, 03-110). Purified RNAs were detected by RT-qPCR. Gene-specific primers are presented in Table 2.

### 2.13. Chromatin Immunoprecipitation

PROMO [27,28] and JASPAR [29] were used to predict transcription factors that could bind to the LINC00663 promoter region. Chromatin immunoprecipitation (ChIP) experiments were performed using a SimpleChIP^®^ Plus Enzymatic Chromatin IP Kit (CST) according to the manufacturer’s instructions. The antibody used for ChIP was 5 μg of anti-FOXA1 (Abcam, ab170933). Purified DNA was detected by RT-qPCR. Primers used for PCR in ChIP experiments were the forward sequence 5′-ACAGAGCCAGGTGAAACAGAG-3′ and the reverse sequence 5′-AACGATTGGGCG-CTCTAAGG-3′.

### 2.14. Animal Studies

Mice experiments were approved by the Institutional Ethics Committee of China Medical University. In a BDL-induced mice liver fibrosis model, 20 Balb/c male mice aged between 8 and 9 weeks were randomly divided into four groups: sham operation (Sham, n = 5); BDL operation (BDL, n = 5); BDL operation treated with injection of pcDNA3.1-NC (BDL + pcDNA, n = 5); and BDL operation treated with injection of pcDNA3.1-LINC00663 (BDL + pcDNA-LIN, n = 5). Each plasmid sample was injected through the tail vein (2 × 10^6^ cells per mouse) after a surgical operation [30]. All mice were killed after 21 days [31]. Liver samples and sera were collected for the analysis of liver function and fibrotic index.

### 2.15. Histology and Immunohistochemistry

Tissue specimens were fixed with 4% paraformaldehyde. For hematoxylin and eosin (HE) and Masson staining, paraffin-embedded tissues were deparaffinized, hydrated, HE or Masson stained and dehydrated. Then, we used a MaxVision^TM^ HRP-polymer anti-mouse/rabbit immunohistochemistry (IHC) Kit (MXB Biotechnologies, Fujian, China) and DAB Kit (20×; MXB Biotechnologies) to perform IHC assays. The primary antibodies used were anti-α-SMA (1:200, CST, #59) and anti-COL1A1 (1:200, Santa Cruz, sc-59772).

### 2.16. Hydroxyproline Assay and Serum ALT and AST

Hydroxyproline in liver samples was measured using hydroxyproline detection kits (Nanjing Jiancheng Biochemical Institute, Nanjing, China) to detect total collagen content. Serum ALT and AST were tested by an automatic analyzer (Hitachi 7600, Hitachi, Tokyo, Japan).

### 2.17. Wound Healing Assay

Scratches were made with a 200 μL pipette tip 48 h post-transfection and growth media was replaced with serum-free DMEM. Cell migration at 0 h and 48 h at the same location were recorded by taking photos. The percentage of wound healing was quantified as (0-h scratch area—48-h scratch area)/0-h scratch area × 100%.

### 2.18. Statistical Analysis

All data were normalized using a Shapiro–Wilk test. Normal distribution data were analyzed using a *t*-test, while non-normal data were analyzed using a Mann–Whitney U test. A two-tailed *p*-value < 0.05 was considered statistically significant.

## 3. Results

### 3.1. Data Preprocessing and Co-Expression Network Construction

We downloaded eight raw data sets from the GEO database. After probe conversion and the removal of duplicate genes, a total of 21,747 genes were obtained. Subsequently, we used analysis of variance to calculate and screen out the top 25% of genes (n = 5437) for co-expression network construction. We chose to define the adjacency matrix β = 10 based on scale-free topology criteria. As a result, the index reached 0.84 (Appendix A). In view of a dynamic tree cutting method, 5437 genes were separated into 15 modules (Appendix A).

### 3.2. Identification of Significant Modules

A darkolivegreen module (r = 0.99, *p* < 0.05) with 1304 genes and darkgrey module (r = −0.89, *p* < 0.05) with 1072 genes were strongly correlated with cell status after the cellular state was introduced into the weighted network (Figure 1A). The following analysis calculated the GS and MM of the darkolivegreen module (r = 0.97, *p* < 0.05, Figure 1B) and the darkgrey module (r = 0.82, *p* < 0.05, Figure 1C). Therefore, these two modules were identified as important modules.

### 3.3. Functional Annotation

Information on 116 key genes is shown in Appendix A. For GO analysis, genes were predominantly rich for extracellular protein matrix, extracellular space, extracellular exosomes, protein binding and protein homodimerization activity (Appendix A). For KEGG analysis, genes were concentrated primarily through ECM–receptor interaction and cell-cycle pathways (Appendix A).

### 3.4. PPI Network Construction and Hub Gene Identification

A PPI network is shown for 116 key genes contained in 116 nodes and 157 edges (Figure 1D). Using the mcode [22] software, we eventually obtained seven genes with degree > 10 (Table 3). Of these, FN attracted our attention due to it being the highest degree. Fibronectin is a macromolecular glycoprotein that constitutes the extracellular matrix and is secreted by HSCs after activation [8]. The pre-mRNA of FN is thought to undergo alternative splicing to produce variants such as extra domain A-FN (EDA-FN), which is crucial in fibrosis [9]. Splicing factor 2 is a clear and found earlier predominant factor in the alternative splicing of FN pre-mRNA, and the level of SF2 expression directly affects the amount of EDA-FN produced [10,11,12]. Therefore, we selected SF2 as a downstream target gene.

### 3.5. Construction and Validation of ceRNA Network

In TGF-β–treated LX-2 cells, both mRNA (Figure 2A) and protein (Figure 2B) levels of α-SMA and COL1A1 were significantly increased, showing that cells were successfully activated. Since the activation levels of the activation markers for 48 h were better than that of activation for 24 h, we used a model of activation for 48 h in the following experiments: We verified the expression of SF2, FN, and EDA-FN in activated LX-2 cells and found that the expression of all three was upregulated in activated LX-2 cells compared to control cells (Figure 2C,D, *p* < 0.05). Through miRNA target prediction websites, we found that hsa-miR-3916 bound to SF2 and that the expression of hsa-miR-3916 was downregulated in activated LX-2 cells compared to control cells (Figure 2E, *p* < 0.05). Similarly, we found that LINC00663 bound to hsa-miR-3916 and the expression of LINC00663 was upregulated in activated LX-2 cells compared to control cells (Figure 2F, *p* < 0.05). Taken together, we constructed and verified a LINC00663/hsa-hsa-miR-3916/SF2-FN ceRNA network. Furthermore, subcellular fractionation analysis showed that LINC00663 were localized both in the cytoplasm and nucleus (Figure 2G).

### 3.6. LINC00663 Regulated the Activation of LX-2 Cells

To study the role of LINC00663 in the activation of LX-2 cells, activated LX-2 cells were transfected with si-LINC00663 and pcDNA-LINC00663. The expression of LINC00663 was decreased in cells of the si-LINC00663 group compared to those of the si-NC group (Figure 3A, *p* < 0.05), and increased in the pcDNA-LIN group compared to the pcDNA group (Figure 3B, *p* < 0.05). The levels of α-SMA and COL1A1 were lower in the si-LINC00663 group compared to the si-NC group (Figure 3C,D, *p* < 0.05) and were higher in the pcDNA-LIN group compared to the pcDNA group (Figure 3E,F, *p* < 0.05). These results showed that LINC00663 regulated the activation of LX-2 cells.

### 3.7. LINC00663 Regulated Epithelial–Mesenchymal Transition and Migration of LX-2 Cells

To study the role of LINC00663 in epithelial–mesenchymal transition (EMT) and migration of LX-2 cells, activated LX-2 cells were transfected with si-LINC00663 and pcDNA-LINC00663. The levels of desmin and vimentin were reduced while that of E-cad was increased by LINC00663 knockdown (Figure 4A, *p* < 0.05). The expression of desmin and vimentin was increased and that of E-cad was reduced when LINC00663 was overexpressed (Figure 4B, *p* < 0.05). To further validate the hypothesis that LINC00663 promoted migration, wound healing assays were employed. As shown in Figure 4C,D, LINC00663 knockdown alleviated migration and overexpression of LINC00663 aggravated migration of LX-2 cells (*p* < 0.05). These data suggested that LINC00663 regulated the EMT and migration of LX-2 cells.

### 3.8. Overexpression of LINC00663 Aggravated BDL-Induced Hepatic Fibrosis In Vivo

To investigate the effect of LINC00663 in BDL-induced hepatic fibrosis, mice were given pcDNA-LINC00663 (pcDNA-LIN) or a negative control (pcDNA). We found that liver tissues treated with pcDNA-LIN had a higher expression of LINC00663 compared with those of the pcDNA group (Figure 5A, *p* < 0.05). Liver hydroxyproline levels (Figure 5B) and alanine aminotransferase and aspartate transaminase serum levels (Table 4) in pcDNA-LIN-injected mice were also increased compared to BDL-induced mice injected with pcDNA. However, the administration of pcDNA-LIN exacerbated BDL-induced hepatic fibrosis, as indicated by macroscopic examination, HE and Masson staining, IHC and WB of α-SMA and COL1A1 (Figure 5C,D). These data suggested that LINC00663 might exacerbate BDL-induced hepatic fibrosis.

### 3.9. LINC00663 Regulated Activation, EMT, and Migration of LX-2 Cells through Hsa-miR-3916

Using RegRNA 2.0, we found two possible binding sites for LINC00663 and hsa-miR-3916. We constructed reporter vectors in which potential binding sites in the sequence for LINC00663 were all or individually mutated (MUT3-LINC00663 was a mutant type of both binding sites; Figure 6A). As expected, cotransfection of MUT3-LINC00663 with a hsa-hsa-miR-3916 mimic did not repress luciferase activity (Figure 6A, *p* < 0.05). To study the role of hsa-hsa-miR-3916, activated LX-2 cells were transfected with a hsa-hsa-miR-3916 mimic or inhibitor. The expression of hsa-hsa-miR-3916 increased in the hsa-hsa-miR-3916 mimic group and decreased in the hsa-hsa-miR-3916 inhibitor group compared to the miR-NC group (Figure 6B, *p* < 0.05). Meanwhile, LINC00663 expression decreased in the hsa-hsa-miR-3916 mimic group and increased in the hsa-hsa-miR-3916 inhibitor group (Figure 6C, *p* < 0.05) indicating that hsa-miR-3916 negatively regulated LINC00663. Similarly, activated LX-2 cells were transfected with si-LINC00663 and pcDNA-LIN; the level of hsa-miR-3916 increased in the si-LINC00663 group and decreased in the pcDNA-LIN group (Figure 6D, *p* < 0.05), explained by LINC00663 negatively regulating hsa-miR-3916. Next, we measured activation and EMT markers. A hsa-miR-3916 inhibitor partially reversed the effect of LINC00663 silencing on the activation and EMT of LX-2 cells (Figure 6E–G, *p* < 0.05), whereas a hsa-miR-3916 mimic partially reversed the effect of LINC00663 overexpression (Appendix A, *p* < 0.05). A wound healing assay showed that a hsa-miR-3916 inhibitor partially reversed the effect of LINC00663 silencing on the migration of LX-2 cells (Figure 6H, *p* < 0.05), whereas a hsa-miR-3916 mimic partially reversed the effect of LINC00663 overexpression (Appendix A, *p* < 0.05).

### 3.10. LINC00663 Regulated SF2-FN Alternative Splicing by Sponging Hsa-miR-3916

Using Targetscan and miRDB, we identified a potential binding site for hsa-miR-3916 and SF2. We constructed a reporter vector with a mutated potential binding site for SF2 sequences (Figure 7A). As expected, co-transfection of MUT-SF2 and a hsa-miR-3916 mimic failed to suppress luciferase activity (Figure 7A, *p* < 0.05). To investigate whether LINC00663 regulated SF2, FN and EDA-FN expression, activated LX-2 cells were transfected with si-LINC00663 and pcDNA-LIN. LINC00663 knockdown reduced SF2 and EDA-FN expression (Figure 7B,C, *p* < 0.05), whereas LINC00663 overexpression promoted SF2 and EDA-FN expression (Figure 7D–E, *p* < 0.05) in activated LX-2 cells; however, neither of these regulated the expression of FN, indicating that LINC00663 regulated SF2-FN alternative splicing (Figure 7D,E, *p* > 0.05). To further demonstrate that LINC00663 regulated SF2-FN alternative splicing by sponging hsa-miR-3916, we performed a rescue assay in activated LX-2 cells. A miR-3916 inhibitor rescued the reduction of SF2 and EDA-FN in cells regulated by LINC00663 (Figure 7F,G, *p* < 0.05), whereas a hsa-miR-3916 mimic abolished the elevation of SF2 and EDA-FN in LINC00663-regulated cells (Appendix A, *p* < 0.05). RNA immunoprecipitation assays were performed in LX-2 cells using Ago2 antibody to investigate the presence of interactions between LINC00663, hsa-miR-3916, and SF2. These three molecules were found to be significantly enriched in the anti-Ago2 group compared to the IgG group (Figure 7H, *p* < 0.05) implying a competitive regulatory relationship between LINC00663, hsa-miR-3916, and SF2.

### 3.11. LINC00663 Regulated by the Transcription Factor FOXA1

We used PROMO and JASPAR to predict the binding site of FOXA1 to the LINC00663 promoter region (Figure 8A). Wild and mutated promotor sequences were constructed in a pGL3-basic vector. A dual-luciferase activity test showed that co-transfection of wild-LINC00663 and pGL3-FOXA1 increased luciferase activity (Figure 8B, *p* < 0.05) illustrating that FOXA1 bound to the promoter region of LINC00663. Furthermore, FOXA1 expression was upregulated in activated LX-2 cells compared to a control group (Figure 8C,D, *p* < 0.05). To investigate the relationship between FOXA1 and LINC00663, activated LX-2 cells were transfected with si-FOXA1 or pEGFP-FOXA1. We found that FOXA1 upregulated LINC00663 (Figure 8E,F, *p* < 0.05). Moreover, we performed ChIP assays in LX-2 cells using FOXA1 antibody. We found that the LINC00663 promoter was significantly enriched using anti-FOXA1 compared to IgG control antibody, suggesting a direct binding relationship between FOXA1 and the LINC00663 promoter region (Figure 8G, *p* < 0.05). Together, these results identified an important regulatory axis whereby LINC00663 that was regulated by FOXA1 sponged hsa-miR-3916 and regulated SF2-FN alternative splicing expression in cholestatic liver fibrosis (Figure 8H).

## 4. Discussion

In recent years, with the widespread recognition of cholestatic liver fibrosis and the gradual increased use of serum anti-mitochondrial antibody, the incidence of PBC and PSC has shown a rapid upward trend. The incidence of PBC in China increased from 2.16/100,000 in 1991 to 2000 to 8.99/100,000 in 2011–2020; PBC cases have increased worldwide [32]. At present, epidemiological data on PSC in China is lacking, but data from Northern Europe and North America have shown that the incidence of PBC has been steadily increasing year after year [33].

The hypothesis that is the basis for ceRNAs is that lncRNAs can affect gene expression by competing with mRNAs for binding to miRNAs [34]. Research reports have highlighted how lncRNAs, such as lncRNA-p21 [35], GAS5 [36] and PVT1 [37], can sponge target miRNAs to influence the activation of HSCs, which may occupy an important regulatory place in the HF process. In the present study, we explored the role of LINC00663 in the progression of HF for the first time. As expected, our data revelated that LINC00663 was overexpressed in activated LX-2 cells, and overexpression of LINC00663 accelerated the progression of HF through sponging miR-3916 to upregulate SF2-FN expression. In addition, previous studies confirmed that LINC00663 promoted the migration abilities of cells by regulating the AKT/mTOR pathway [38]. This is consistent with our study that LINC00663 overexpression can promote the migration of LX-2 cells. The mechanism of hsa-miR-3916 in liver fibrosis has not been reported yet. In prostate cancer, high expression of hsa-miR-3916 reduced the protein level of pyruvate dehydrogenase kinase 1, thereby reducing the risk of prostate cancer [39]. However, in Conjunctival Melanoma patients, upregulation of hsa-miR-3916 was associated with a higher risk of local recurrence [40].

FN is a macromolecular glycoprotein constituting the extracellular matrix, secreted by HSCs after activation and is an important molecule for the formation of HF [8]. As an important substrate of TGF-β, the alternative spliced form of FN, EDA-FN participates in the induction of fibroblast proliferation and differentiation, EMT, fibroblast extracellular matrix production and tissue fibrosis [41,42]. The interaction of TGF-β with EDA-FN promotes the transformation of fibroblasts into α-SMA-expressing myofibroblasts and promotes fibrosis [43,44]. Splicing factor 2 is a kind of serine/arginine rich protein (SR protein) [45]. The capability of SR proteins is to facilitate selection of splice sites by binding to exon splicing enhancers [45]. Just as SF2 is a predominant factor in alternative splicing of FN pre-mRNA, the level of SF2 directly affects the production of EDA-FN [10,11]. SF2 can control alternative splicing of the tumor suppressor to disable their function. Therefore, it also is considered to be an oncoprotein [46].

Long non-coding RNAs are often regulated by transcription factors and are mainly transcribed by RNA polymerase II and 5′-capped, and polyadenylated in the same manner as protein coding mRNAs [31,47]. Forkhead box A is a transcribed protein that has regulatory effects on liver metabolism and development, and is a major transcription factor in epithelial lineage differentiation [48,49,50]. In breast cancer cells, FOXA1 and DSCAM-AS1 form a positive feedback loop to promote cancer cell proliferation [51]. Through our study, we found that FOXA1 induced the upregulation of LINC00663 in LX-2 cells and had a strong affinity to the promoter of LINC00663.

We exhaustively elucidated the relationships between FOXA1/LINC00663/hsa-miR-3916/SF2-FN in the modulation of the progression of cholestatic liver fibrosis. However, limitations existed in our study. We did not verify the expression of related molecules in clinical samples, so we could not evaluate the diagnostic value of LINC00663 as a biomarker, in human hepatic stellate LX-2 cells and in bile duct-ligated cholestasis mice.

## 5. Conclusions

In summary, we elaborated the fibrotic effects of LINC00663 in human hepatic stellate LX-2 cells and in bile duct-ligated cholestasis mice. We established a FOXA1/LINC00663/hsa-miR-3916/SF2-FN axis that provided a potential target for the diagnosis and targeted therapy of cholestatic liver fibrosis.

## Figures and Tables

**Figure 1 cells-12-00215-f001:**
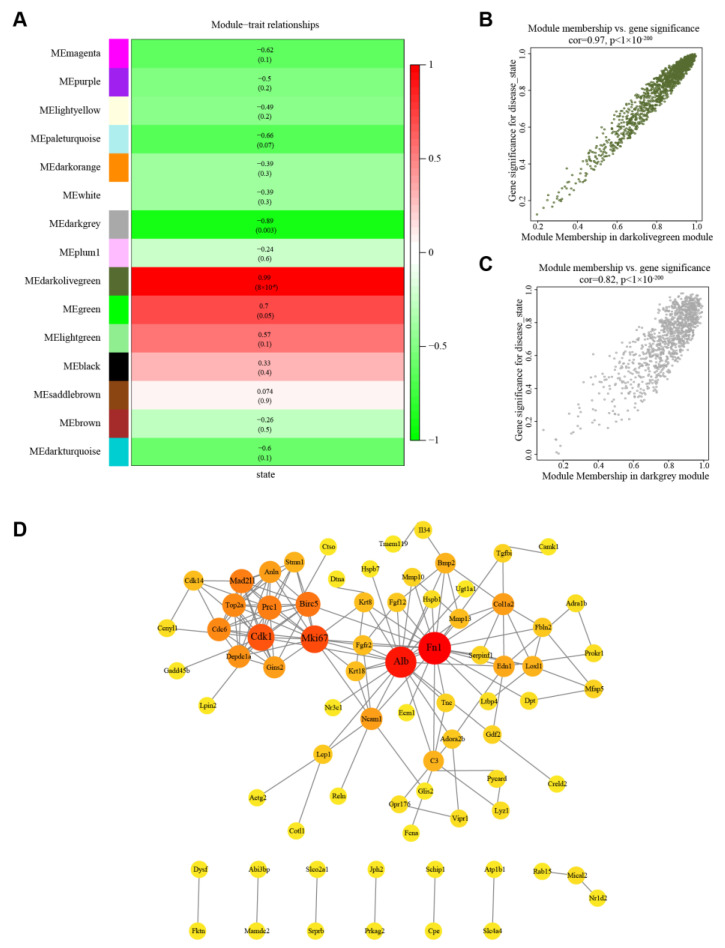
Identification of significant modules and key genes. (**A**) The relationship between modules and cell states. Cell states included quiescent and activated HSCs. (**B**) A scatter plot of darkolivegreen module genes and cell states. (**C**) A scatter plot of darkgrey module genes and cell states. (**D**) A PPI network of key genes using the online tool, STRING. HSCs, hepatic stellate cells; PPI, protein–protein interaction network.

**Figure 2 cells-12-00215-f002:**
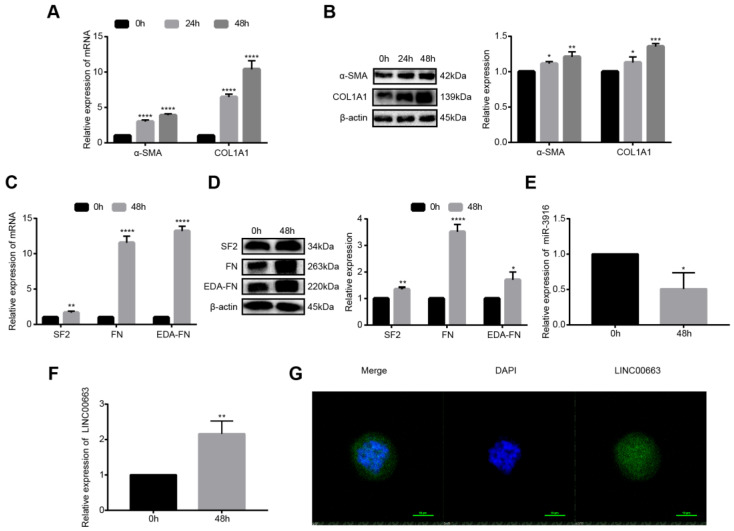
Validation of ceRNA network in activated LX-2 cells. Relative expression of α-SMA and COL1A1 in LX-2 cells treated with 10 ng/mL TGF-β for 24 h or 48 h detected by (**A**) RT-qPCR and (**B**) WB. Relative expression of SF2, FN and EDA-FN in LX-2 cells treated with 10 ng/mL TGF-β1 for 48 h detected by (**C**) RT-qPCR and (**D**) WB. (**E**) Hsa-miR-3916 was downregulated in activated LX-2 cells. (**F**) LINC00663 was upregulated in activated LX-2 cells. (**G**) RNA FISH was performed to determine the location of endogenous LINC00663 (green) in LX-2 cells. Nuclei were counterstained with DAPI (blue). Scale bar = 10 μm. * *p* < 0.05, ** *p* < 0.01, *** *p* < 0.001, **** *p* < 0.0001. Each experiment was repeated three times and similar results were obtained. ceRNA, competing endogenous RNA; COL1A1, collagen type I alpha 1 chain; EDA-FN, extra domain A of fibronectin; FISH, fluorescence in situ hybridization; FN, fibronectin; HSCs, hepatic stellate cells; RT-qPCR, real-time quantitative PCR; α-SMA, α-smooth muscle actin; SF2, splicing factor 2; TGF-β, transforming growth factor-β; WB, western blot.

**Figure 3 cells-12-00215-f003:**
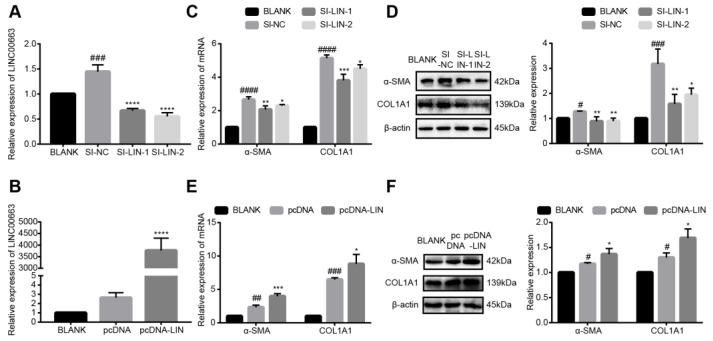
LINC00663 regulated activation of LX-2 cells. LINC00663 expression in LX-2 cells transfected with (**A**) si-LINC00663 (SI-LIN-1 and SI-LIN-2) and (**B**) pcDNA-LINC00663 (pcDNA-LIN). Relative expression of α-SMA and COL1A1 in LX-2 cells transfected with si-LINC00663 detected by (**C**) RT-qPCR and (**D**) WB. Relative expression of α-SMA and COL1A1 in LX-2 cells transfected with pcDNA-LIN detected by (**E**) RT-qPCR and (**F**) WB. * *p* < 0.05, ** *p* < 0.01, *** *p* < 0.001, **** *p* < 0.0001. # *p* < 0.05, ## *p* < 0.01, ### *p* < 0.001, #### *p* < 0.0001. * vs. NC, # vs. BLANK. Each experiment was repeated three times and similar results were obtained. COL1A1, collagen type I alpha 1 chain; HSCs, hepatic stellate cells; RT-qPCR, real-time quantitative PCR; si, small interfering RNA; α-SMA, α-smooth muscle actin; WB, western blot.

**Figure 4 cells-12-00215-f004:**
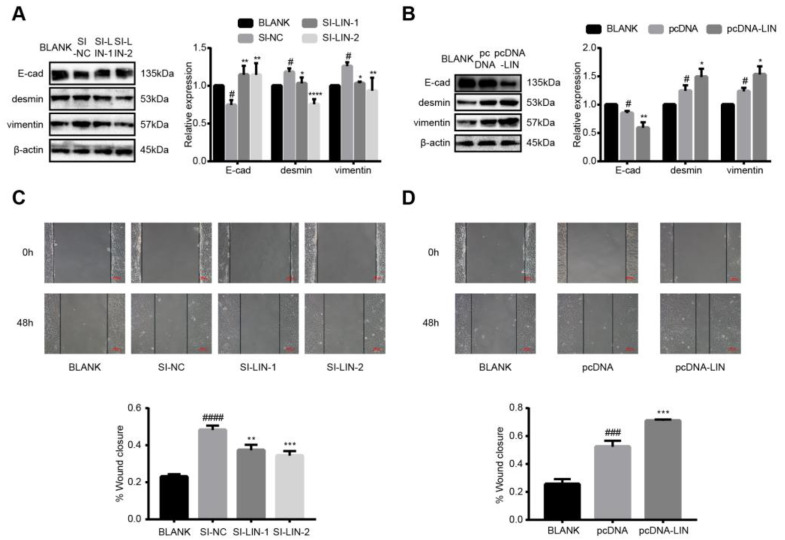
LINC00663 regulated EMT and migration of LX-2 cells. Relative expression of E-cadherin, desmin, and vimentin in LX-2 cells transfected with (**A**) si-LINC00663 (SI-LIN-1 and SI-LIN-2) and (**B**) pcDNA-LINC00663 (pcDNA-LIN) detected by WB. The effect of LINC00663 on cell migration in LX-2 cells transfected with (**C**) si-LINC00663 and (**D**) pcDNA-LIN detected by wound healing assay. Scale bar = 100 μm. * *p* < 0.05, ** *p* < 0.01, *** *p* < 0.001, **** *p* < 0.0001. # *p* < 0.05, ### *p* < 0.001, #### *p* < 0.0001. * vs. NC, # vs. BLANK. Each experiment was repeated three times and similar results were obtained. EMT, epithelial–mesenchymal transition; HSCs, hepatic stellate cells; si, small interfering RNA; WB, western blot.

**Figure 5 cells-12-00215-f005:**
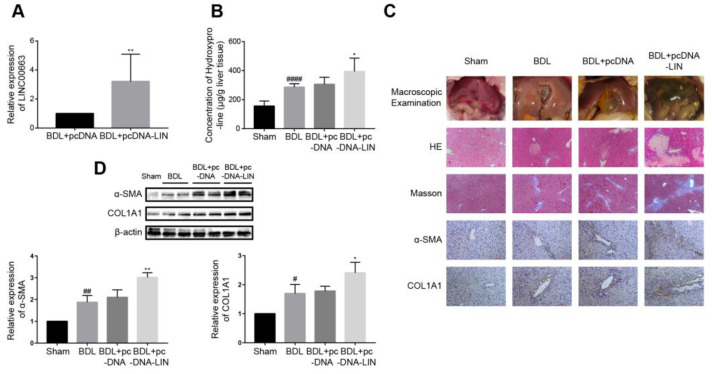
Overexpression of LINC00663 aggravated BDL-induced hepatic fibrosis in vivo. (**A**) RT-qPCR analysis showed LINC00663 expression in the liver of BDL mice administered pcDNA-LINC00663 (pcDNA-LIN) or negative control (NC) (n = 7 per group). (**B**) Quantification of hepatic hydroxyproline content. The data were expressed as hydroxyproline (μg) per liver wet weight (g) (n = 7 per group). (**C**) Liver fibrosis was evaluated by macroscopic examination, HE staining, Masson’s staining and IHC in sham or BDL mice administered pcDNA-LIN or NC. Scale bars, 100 μm for HE (objective, ×10), Masson’s staining (objective, ×10) and IHC (objective, ×20). (**D**) Relative expression of α-SMA and COL1A1 in BDL mice transfected with pcDNA-LIN detected by WB. * *p* < 0.05, ** *p* < 0.01. # *p* < 0.05, ## *p* < 0.01, #### *p* < 0.0001. * vs. BDL+ pcDNA, # vs. Sham. Each experiment was repeated three times and similar results were obtained. BDL, bile duct ligation; HE, hematoxylin and eosin; IHC, immunohistochemistry; NC, negative control; RT-qPCR, real-time quantitative PCR.

**Figure 6 cells-12-00215-f006:**
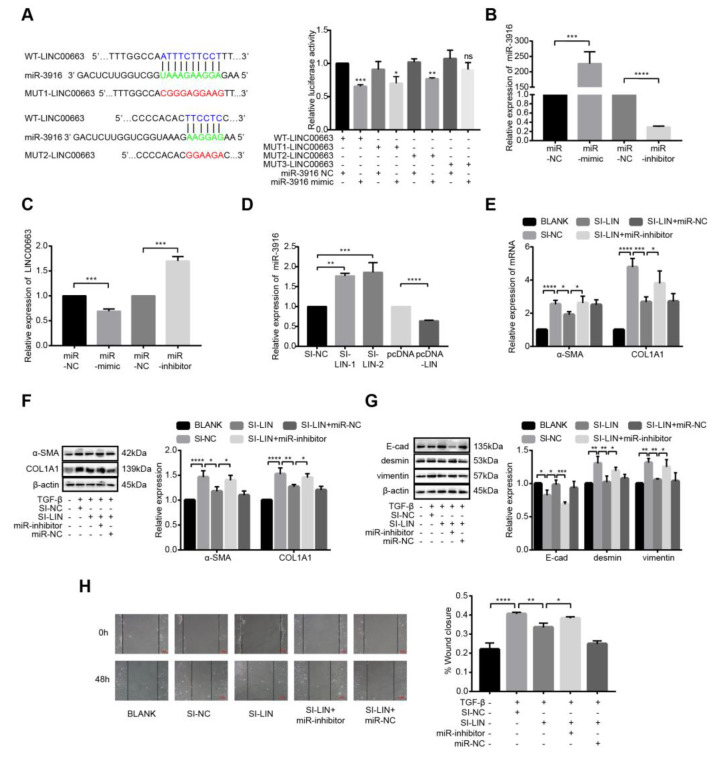
LINC00663 regulated activation, EMT, and migration of LX-2 cells through hsa-miR-3916. (**A**) A schematic diagram showing the sequences of wild-type and mutant LINC00663 with hsa-miR-3916. A dual-luciferase reporter gene assay was performed to identify the interaction between LINC00663 and hsa-miR-3916. The expression of hsa-miR-3916 (**B**) and LINC00663 (**C**) in LX-2 cells transfected with hsa-miR-3916 mimic (miR-mimic) and hsa-miR-3916 inhibitor (miR-inhibitor). (**D**) The expression of hsa-miR-3916 in LX-2 cells transfected with si-LINC00663 and pcDNA-LINC00663 (pcDNA-LIN). Relative expression of α-SMA and COL1A1 in LX-2 cells transfected with si-LIN, miR-inhibitor or with both detected by (**E**) RT-qPCR and (**F**) WB. (**G**) Relative expression of EMT proteins in LX-2 cells transfected with si-LIN, miR-inhibitor or with both detected by WB. (**H**) The effect of LINC00663 and hsa-miR-3916 on cell migration by LX-2 cells transfected with si-LIN, miR-inhibitor, or with both detected by wound healing assay. Scale bar = 100 μm. * *p* < 0.05, ** *p* < 0.01, *** *p* < 0.001, **** *p* < 0.0001. Each experiment was repeated three times and similar results were obtained. COL1A1, collagen type I alpha 1 chain; EMT, epithelial–mesenchymal transition; HSCs, hepatic stellate cells; miR, microRNA; siRNA, small interfering RNA; α-SMA, α-smooth muscle actin; WB, western blot.

**Figure 7 cells-12-00215-f007:**
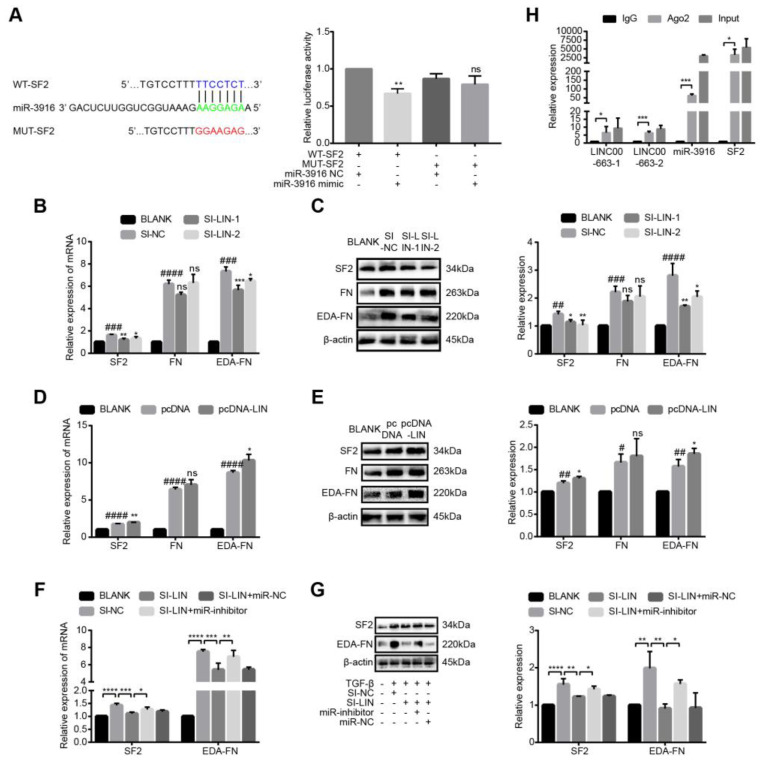
LINC00663 regulated SF2-FN alternative splicing by sponging hsa-miR-3916. (**A**) A schematic diagram showing the sequences of wild-type and mutant SF2 with hsa-miR-3916. A dual-luciferase reporter gene assay was performed to identify the interaction between SF2 and hsa-miR-3916. Relative expression of SF2, FN, and EDA-FN in LX-2 cells transfected with si-LINC00663 detected by (**B**) RT-qPCR and (**C**) WB. Relative expression of SF2, FN, and EDA-FN in LX-2 cells transfected with pcDNA-LINC00663 detected by (**D**) RT-qPCR and (**E**) WB. Relative expression of SF2 and EDA-FN in si-LIN, miR-inhibitor, or both transfected LX-2 cells detected by (**F**) RT-qPCR and (**G**) WB. (**H**) RIP experiment with anti-Ago2, IgG as a negative control, and input as a positive control from LX-2 cell extracts using RT-qPCR. * *p* < 0.05, ** *p* < 0.01, *** *p* < 0.001, **** *p* < 0.0001. # *p* < 0.05, ## *p* < 0.01, ### *p* < 0.001, #### *p* < 0.0001. * vs. NC, # vs. BLANK. ns vs. BLANK, not statistically significant. Each experiment was repeated three times and similar results were obtained. EDA-FN, extra domain A of fibronectin, FN, fibronectin; RIP, RNA immunoprecipitation; RT-qPCR, real-time quantitative PCR; SF2, splicing factor 2; siRNA, small interfering RNA; WB, western blot.

**Figure 8 cells-12-00215-f008:**
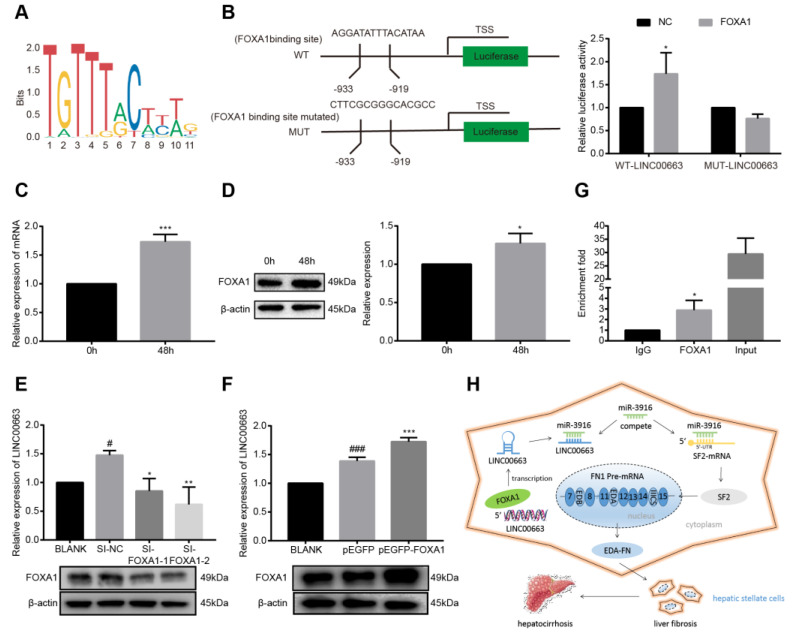
LINC00663 was regulated by FOXA1 transcription factor. (**A**) Bioinformatics predicted binding sites between FOXA1 and LINC00663. (**B**) A schematic diagram shows sequences of wild-type and mutant LINC00663 with FOXA1. A dual-luciferase reporter gene assay was performed to identify the interaction between LINC00663 and FOXA1. FOXA1 was upregulated in activated LX-2 cells as detected by (**C**) RT-qPCR and (**D**) WB. Relative expression of LINC00663 in LX-2 cells transfected with (**E**) si-FOXA1 and (**F**) pEGFP-FOXA1 detected by RT-qPCR. (**G**) ChIP experiment with FOXA1, IgG as negative control, and input as a positive control from LX-2 cell extracts using RT-qPCR. (**H**) Schematic representation of FOXA1/LINC00663/hsa-miR-3916/SF2-FN regulatory axis. * *p* < 0.05, ** *p* < 0.01, *** *p* < 0.001. # *p* < 0.05, ### *p* < 0.001. * vs. NC, # vs. BLANK. Each experiment was repeated three times and similar results were obtained. ChIP, chromatin immunoprecipitation; FN, fibronectin; FOXA1, forkhead box A1; HSCs, hepatic stellate cells; RT-qPCR, real-time quantitative PCR; SF2, splicing factor 2; siRNA, small interfering RNA; WB, western blot.

**Table 1 cells-12-00215-t001:** Forward and reverse primers for real-time PCR.

Genes	Sequence (5′-3′)
LINC00663	F: GCAGCGATGATGACCGTAA
	R: GGTGTTCTCAACTCATCCAC
α-SMA	F: GGCAAGTGATCACCATCGGA
	R: GTGGTTTCATGGATGCCAGC
COL1A1	F: GAGGGCCAAGACGAAGACATC
	R: CAGATCACGTCATCGCACAAC
FN	F: GGTGACACTTATGAGCGTCCTAAA
	R: AACATGTAACCACCAGTCTCATGTG
EDA-FN	F: GGAGAGAGTCAGCCTCTGGTTCAG
	R: TGTCCACTGGGCGCTCAGGCTTGTG
SF2	F: TGTCGGGAGGTGGTGTGATTCGT
	R: CTTGGTTCGGATGTCTGGAGGTAAGTT
β-actin	F: AACTGGGACGACATGGAGAAAA
	R: GGATAGCACAGCCTGGATAGCAA
hsa-miR-3916	F: CGAAGAGGAAGAAATGGCTGGT
	R: AGTGCAGGGTCCGAGGTATT
	RT primer: GTCGTATCCAGTGCAGGGTCCGAGGTATTCGCACTGGATACGACCTGAGA
U6	F: CTCGCTTCGGCAGCACA
	R: AACGCTTCACGAATTTGCGT

F, forward primer; R, reverse primer; α-SMA, α-smooth muscle actin; COL1A1, collagen type I alpha 1 chain; EDA-FN, extra domain A of fibronectin, FN, fibronectin; SF2, splicing factor 2; RT, reverse transcription.

**Table 2 cells-12-00215-t002:** Forward and reverse primers for RIP real-time PCR.

Targets	Sequence (5′-3′)
LINC00663-1	F: TTGGACTTGTGTAGGCTTGTAG
	R: GGTATAGGCATTGGATCAACATTC
LINC00663-2	F: GTGAGAGAGTTTGCATGACATCT
	R: TGAGCAGGAGGAAGTGTG
SF2	F: CAATTGGCTATTGCCGTTT
	R: CTGATAGACCCAGTTAGAATCT

RIP, RNA immunoprecipitation; F, forward primer; R, reverse primer; SF2, splicing factor 2.

**Table 3 cells-12-00215-t003:** Full names and degrees of hub genes.

Hub Gene	Full Name	Degree
FN	fibronectin	23
Alb	albumin	21
Mki67	marker of proliferation Ki-67	16
CDK1	cyclin dependent kinase 1	15
Birc5	baculoviral IAP repeat containing 5	12
Mad2l1	mitotic arrest deficient 2 like 1	11
Prc1	protein regulator of cytokinesis 1	11

**Table 4 cells-12-00215-t004:** Serum ALT and AST levels of mice in each group (mean ± SD, n = 7).

Group	ALT (U/L)	AST (U/L)
Sham	75.0 ± 34.43	182.4 ± 74.56
BDL	284.3 ± 80.21 ^####^	427.9 ± 57.24 ^####^
BDL + pcDNA	343.9 ± 78.34	488.6 ± 69.99
BDL + pcDNA-LIN	477.3 ± 139.2 *	594.1 ± 101.1 *

ALT, alanine aminotransferase; AST, aspartate transaminase; BDL, bile duct ligation; ^####^
*p* < 0.0001 vs. Sham, * *p* < 0.05 vs. BDL+ pcDNA.

## Data Availability

The data involved in this study were available through NCBI GEO (GSE34640).

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
