# Peer review of "The Fibrotic Effects of LINC00663 in Human Hepatic Stellate LX-2 Cells and in Bile Duct-Ligated Cholestasis Mice Are Mediated through the Splicing Factor 2-Fibronectin"

_cells, 2023, doi:10.3390/cells12020215_

Round 1

Reviewer 1 Report

1.Are there other gene expression directly affects the amount of EDA-FN produced?Why do you chose SF2 amoing these gene?

2. Please explain more about “hsa-miR-3916”and SF2in discussion, including thier function or mechanism mentioned in previous studies.

Author Response

Dear Reviewer,

Thank you very much for your attention and for the reviewer’s evaluation and comments on our paper. Point-by-point responses to each of the comments of the reviewer are listed in the PDF. Please see the attachment. Thank you.

Correspondence Author: Hui Kang

Reviewer 2 Report

The authors delineated a signaling axis (FOXA1/LINC00663/hsa-miR-3916/SF2-FN) in liver fibrosis. Overall, the experimental design is sound and the data appear to support the conclusion. The key role of lINC00663 in fibrosis was confirmed in both cell and animal models. The signaling axis is novel as well.  The reviewer has the following comments for the authors:

1. The authors performed an analysis of GEO data sets to identify key genes and nodules involved in liver fibrosis. The authors clarified why they chose to focus on FN and SF2, which subsequently leads to the discovery of hsa-miR-3916 as well as LINC00663. However, it is unclear why the authors chose to focus most of the efforts on LINC00663 instead of SF2 or hsa-miR-3916?  The logical flow of the article will be improved if this is more clearly articulated.

2. What is the cell type specificity of this signaling axis? If we modulate this signaling axis in skin fibroblasts, would it give us the same results?

3. For the mice study, it seems the authors just injected the plasmids systematically. Is this going to result in efficient transfection? Is there any prior literature to support this methodology? What there the systematic effects of such an injection? 

4. Along the same line, for the mice study, the quantitative parameters used to measure fibrosis seem very limited. It would be helpful to perform WB analysis of all the key fibrosis proteins using the liver samples.

Minor:

1. It is very hard to discern the scale bar in Fig.2G.

2. It will be helpful if it is better explained why WGCNA was used. what about starting from DEGs like most papers do?

Author Response

(The authors gave the same response as above.)

Round 2

Reviewer 2 Report

The reviewer has no additional comments.